# Understanding early HIV-1 rebound dynamics following antiretroviral therapy interruption: The importance of effector cell expansion

Tin Phan[1], Jessica M. Conway[2,3], Nicole Pagane[4,5], Jasmine Kreig[1], Narmada Sambaturu[1], Sarafa Iyaniwura[1], Jonathan Z. Li[6], Ruy M. Ribeiro[1], Ruian Ke[1]*, Alan S. Perelson[1,7]*

1 Theoretical Biology and Biophysics, Los Alamos National Laboratory, Los Alamos, New Mexico, United States of America, 2 Department of Mathematics, Pennsylvania State University, College Township, Pennsylvania, United States of America, 3 Department of Biology, Pennsylvania State University, College Township, Pennsylvania, United States of America, 4 Program in Computational and Systems Biology, Massachusetts Institute of Technology; Cambridge, Massachusetts, United States of America, 5 Ragon Institute of MGH, MIT, and Harvard; Cambridge, Massachusetts, United States of America, 6 Department of Medicine, Division of Infectious Diseases, Brigham and Women's Hospital, Harvard Medical School, Boston, Massachusetts, United States of America, 7 Santa Fe Institute, Santa Fe, New Mexico, United States of America

* rke@lanl.gov (RK); asp@lanl.gov (ASP)

**Data Availability Statement:** The de-identified viral load data is available in the Supplementary Information.

## Abstract

Most people living with HIV-1 experience rapid viral rebound once antiretroviral therapy is interrupted; however, a small fraction remain in viral remission for an extended duration. Understanding the factors that determine whether viral rebound is likely after treatment interruption can enable the development of optimal treatment regimens and therapeutic interventions to potentially achieve a functional cure for HIV-1. We built upon the theoretical framework proposed by Conway and Perelson to construct dynamic models of virus-immune interactions to study factors that influence viral rebound dynamics. We evaluated these models using viral load data from 24 individuals following antiretroviral therapy interruption. The best-performing model accurately captures the heterogeneity of viral dynamics and highlights the importance of the effector cell expansion rate. Our results show that post-treatment controllers and non-controllers can be distinguished based on the effector cell expansion rate in our models. Furthermore, these results demonstrate the potential of using dynamic models incorporating an effector cell response to understand early viral rebound dynamics post-antiretroviral therapy interruption.

## Author summary

Most people living with HIV-1 experience rapid viral rebound once antiretroviral therapy is interrupted; however, a small fraction remain in viral remission for an extended duration. The factors that determine viral rebound dynamics after treatment interruption are not well understood. In this study, we built upon a previous theoretical framework to construct dynamic models of virus-immune interactions to study factors that influence viral rebound dynamics. We evaluated these models using viral load data from 24 individuals

**Funding:** This work was performed under the auspices of the US Dept. of Energy under contract 89233218CNA000001 and supported by NIH grants R01-AI152703 (RK), R01-AI028433 (ASP), R01-OD011095 (ASP, JMC), P01-AI169615 (ASP), UM1-AI164561 (RMR), R21-AI143443 (JMC), R01-AI150396 (JZL), P30-AI060354 (JZL), and Los Alamos National Laboratory LDRD 20220791PRD2 (TP), LDRD 20230853PRD2 (JK), and 20210959PRD3 (NS). JMC also acknowledges the support of the National Science Foundation (grant no. DMS-1714654). NP is supported by the U.S. Department of Energy, Office of Science, Office of Advanced Scientific Computing Research, Department of Energy Computational Science Graduate Fellowship under Award Number DE-SC0022158. Research reported in this publication was also supported by the National Institute of Allergy and Infectious Diseases of the National Institutes of Health under Award Number UM1 AI068634, UM1 AI068636, and UM1 AI106701. The content is solely the responsibility of the authors and does not necessarily represent the official views of the National Institutes of Health. The funders had no role in study design, data collection and analysis, decision to publish, or preparation of the manuscript.

**Competing interests:** JMC is a consultant for Excision Biotherapeutics and Merck. The other authors declare no competing interest.

following antiretroviral therapy interruption. The best-performing model accurately captures the heterogeneity of viral dynamics and robustly shows that people who remain in viral remission for an extended duration after treatment interruption have a higher effector cell expansion rate compared to those who rapidly rebound. These results demonstrate the potential of using viral dynamic models incorporating an effector cell response to understand early viral rebound dynamics post-antiretroviral therapy interruption.

## Introduction

Antiretroviral therapy (ART) is successful at suppressing human immunodeficiency virus-1 (HIV-1) below the limit of detection in people with HIV-1 (PWH). However, ART does not completely eliminate HIV-1 due to the presence of latently infected cells, which have low or no HIV-1 gene expression, making it difficult for the immune system to recognize and eliminate them [1]. While continual use of ART robustly controls HIV-1, there are a number of issues with long-term ART, ranging from how well PWH adhere to treatment [2–5] to the potential side effects of taking ART long term [6], such as chronic inflammation and HIV-associated neurocognitive disorders [7–10]. Thus, durable ART-free virological control remains a major goal.

Due to the persistence of the latent reservoir [11–18], if ART is interrupted, latently infected cells that reactivate can lead to viral rebound. In fact, following analytical treatment interruption (ATI) of ART, most persons with HIV-1, including those with an extremely small latent reservoir, experience viral rebound typically within weeks [19–23]. But in rare cases, some individuals maintained low viral loads for an extended duration of months to years [16,22,24,25]. Individuals, who experience rapid viral rebound, are generally referred to as non-controllers (NC), while those who control are termed post-treatment controllers (PTC). The generation of post-treatment control appears to be affected by external factors such as the timing of ART initiation [17,22,25–28], and in human and nonhuman primate models by viral suppression mediated by CD8+ cells [29–34]. Thus, understanding the factors associated with PTC can suggest a clinical path towards durable control in PWH.

Several models have been proposed to explain the mechanisms behind viral rebound following ATI. The studies by Hill et al. [35,36], Pinkevych et al. [37,38], Fennessey et al. [39], and van Dorp et al. [40] hypothesized rebound as a stochastic process due to the reactivation of latently infected cells that release virus and initiate a chain of other successful infection events. Yet, in PTC, viral load is kept at a low level despite a large reservoir size in some people. This suggests that other factors, such as the immune response, also play important roles in determining rebound dynamics of HIV-1 [41–48]. For this reason, Conway and Perelson [49] proposed a model of viral rebound considering both the latent reservoir and immune response dynamics. Their model demonstrated the combined impact of the immune response and the size of the latent reservoir on HIV-1 dynamics post-ATI and lays the foundation for this study.

Some modeling studies have also estimated the distribution of time to rebound. For instance, Conway et al. [50,51] utilized personal biomarkers including HIV-1 reservoir quantification and cell associated HIV-1 RNA to look at the distribution of time to rebound and fit a model to viral rebound data from a collection of ACTG (Advancing Clinical Therapeutics Globally for HIV/AIDS, formerly known as the AIDS Clinical Trials Group) ATI studies [16]. Others studied mathematical and statistical models to fit rebound data of HIV-1 and simian immunodeficiency virus (SIV) [50,52–54], including studies in which additional treatments with monoclonal antibodies or immune stimulants were given prior to ATI [31,55,56]. While these attempts provide valuable insights into the rebound dynamics post-ATI, they either did

not aim to establish the biological factors that distinguish PTC and NC or did not fit to individual viral rebound data. In this work, our objective is to examine whether mechanistic models with virus-immune interactions can accurately capture the heterogeneity in viral rebound dynamics and identify potential mechanisms that distinguish the outcomes post-ATI.

## Methods

### Ethics statement

This research was approved by the Los Alamos National Laboratory Human Subjects Research Review Board (HSRRB). Patients' consent was not required because this is a retrospective study using data from clinical trials reported by Sharaf et al. [57].

### Data

We analyze the data of 24 PWH (9 PTC and 15 NC) from Sharaf et al. [57] for whom we have longitudinal viral load data. The PTC participants were identified from several clinical studies in the ACTG [58–62]. In Sharaf et al., PTCs were defined as individuals who maintained a viral load of less than 400 HIV-1 RNA copies/mL for at least 24 weeks post ATI and where short-term increases ($\leq 2$ viral load measurements) of over 400 HIV-1 RNA copies/mL were not exclusionary. Non-controllers were defined as individuals who did not meet the PTC definition. We chose to analyze the data between 0- and 52-weeks post ATI to focus on the period during which the viral load data is often used to classify PTC and NC in clinical settings. Note that when non-controllers rebounded, they were often put back on ART, which terminated their data sets.

### Mathematical models

Conway and Perelson [49] introduced the following model of viral dynamics to study the phenomenon of PTC:

$$\frac{dT}{dt} = \lambda_T - d_T T - \beta VT$$

$$\frac{dI}{dt} = (1 - f_L)\beta VT + aL - \delta I - mEI$$

$$\frac{dL}{dt} = f_L \beta VT - d_L L - aL + \rho L$$

$$\frac{dE}{dt} = \lambda_E + \frac{bEI}{K_B + I} - \frac{dEI}{K_D + I} - d_E E$$

$$\frac{dV}{dt} = pI - cV. \tag{1}$$

This model (Eq 1) assumes that target cells ($T$) are produced at constant rate $\lambda_T$ and die at per capita rate $d_T$, respectively. Virus (V) infects target cells with rate constant $\beta$ and is cleared at per capita rate $c$. The infection leads to productively infected cells ($I$) at a rate $1$-$f_L$ $\beta VT$, where $f_L$ and $(1$-$f_L)$ are the fractions of infections that lead to latently and productively infected cells, respectively. Infected cells release viruses at per capita rate $p$ and die at per capita rate $\delta$ by viral cytopathic effects. Latently infected cells ($L$) proliferate and decay at per capita rates $\rho$

and $d_L$, respectively, and can reactivate at per capita rate $a$ to become productively infected cells. In the absence of viral infection, effector cells ($E$), such as CD8+ T-cells and natural killer (NK) cells, are produced at rate $\lambda_E$ and die at per capita rate $d_E$. During viral infection, the effector cells recognize infected cells, which stimulates its population to expand. Conway and Perelson modeled this process using the infected-cell dependent growth term $\frac{bEI}{K_B+I}$, where $b$ is the maximal growth rate and $K_B$ is the density of infected cells required for the effector cell growth rate to be at its half-maximal rate. Note if $K_B$ is very large, then a large number of infected cells will be needed to activate the expansion of effector cells, and this will lead to a less effective effector cell response.

The effector cells kill productively infected cells with rate constant $m$, or more precisely, a second order reaction rate constant $m$, and can become exhausted at maximal rate $d$. Effector exhaustion is induced by the upregulation of inhibitory receptors during effector expansion that ultimately abrogate the expansion and other functional responses. Chronic infections and cancers typically lead to effector cell exhaustion, which occurs through metabolic and epigenetic reprogramming of the effector cells that incrementally make them worse at proliferating, surviving, and performing effector functions, such as killing and releasing diffusible molecules [63–65]. Conway and Perelson modeled the rate of exhaustion with a similar infected-cell dependent functional response term $\frac{dEI}{K_D+I}$. These functional response terms had previously been suggested by Bonhoeffer et al. [66].

The model formulation assumes a fraction, $f_L$, of infections lead to the production of latently infected cells. The homeostatic proliferation of latently infected cells at per capita rate $\rho$ follows from prior work [67–70] and is supported by experimental evidence showing that latent cells can proliferate without activating (or expressing viral signals) [18,71–74].

*A Simplified Model.* We formulated variations of the Conway and Perelson model by altering the model assumptions on the dynamics of latent cells and the effector cells (Section A in S1 Text). In particular, without effector cell exhaustion, we have Simplified Model 1:

$$\frac{dT}{dt} = \lambda_T - d_T T - \beta VT$$

$$\frac{dI}{dt} = (1 - f_L)\beta VT + aL - \delta I - mEI$$

$$\frac{dL}{dt} = f_L \beta VT - d_L L - aL + \rho L$$

$$\frac{dE}{dt} = \lambda_E + \frac{bEI}{K_B + I} - d_E E$$

$$\frac{dV}{dt} = pI - cV. \tag{2}$$

For further simplification, we assume that the number of viruses is in quasi-steady state with the number of infected cells in all models including the Conway and Perelson model, so that $V = \frac{p}{c}I$, as in prior work [75,76]. The initial time ($t = 0$) is when the individuals are taken off ART. We let $T(0) = \frac{\lambda_T}{d_T}, I(0) = \frac{aL(0)}{\delta}, E(0) = \frac{\lambda_E}{d_E}$, and $V(0) = \frac{p}{c}I(0)$, which are approximations of the steady states during ART assuming a small latent reservoir at the time of ATI, that ART was highly effective and suppressed all new infections, and thus that productively infected cells were only generated via activation of latently infected cells. We also studied other models

given in Section A in S1 Text. In models without explicit latently infected cell dynamics, we assume the reservoir size does not vary significantly during the first year post-ATI or during the off-ART duration for NCs, the time period we study. This implies $L(t) = L(0)$. Whether the viral load rebounds or not post-ATI is determined by both the immune response and the latent reservoir size. The influence of the reservoir size at the time of ATI has been studied before [36,49], thus we choose to examine whether the immune response—without variation in the latent reservoir size—can capture the viral rebound dynamics for this cohort of participants.

The parameters that we fit, and the fixed parameters are given in Table 1. $f_L$ is fixed to $10^{-6}$, meaning only 0.0001% of infections result in latent cell production [77]; however, increasing $f_L$ up to 10-fold, has little effect on the model dynamics. The assumed size of the latent reservoir was chosen as 1 cell/ml based on estimates of the number of replication competent latently infected cells per million resting CD4 T cells determined by the quantitative viral outgrowth assay [78,79] and assuming patients on long-term ART have close to a million CD4+ T cells per ml. The net rate of change of the latent cell population $d_L - a + \rho$ is such that the expected half-life ($t_{1/2}$) of the reservoir is 44 months [14,15]. We set $d_L = 0.004$ per day [68], and $a = 0.001$ [49], which gives $\rho \approx 0.0045$ per day. We remark that the stability of the latent reservoir is due to the combined effect of proliferation, death, and activation; however, these parameter values are only educated guesses. Furthermore, recent data has suggested that due to proliferation, the reservoir might stop decaying [18].

## Data fitting

We used a nonlinear mixed effects modeling approach (software Monolix 2023R1, Lixoft, SA, Antony, France) to fit Simplified Model 1 and other variants presented in Section A in S1 Text

**Table 1. Model parameters.** The double dashed line separates parameters that we fit (above) and fixed (below). References for the fitting parameters are provided for comparison purposes.

| Parameter | Meaning | Unit | |
|---|---|---|---|
| $\beta$ | Infection rate | $mL$ ($HIV\ RNA\ copies)^{-1}\ day^{-1}$ | [80,81] |
| $\lambda_E$ | Production rate of effector cells | $cells\ mL^{-1}\ day^{-1}$ | [49] |
| $m$ | Effector cell killing rate constant | $mL\ cells^{-1}\ day^{-1}$ | [49] |
| $b$ | Maximum effector cell growth rate | $day^{-1}$ | [82] |
| $K_B$ | Density of infected cells needed to reach a half-maximal effector cell growth rate | $cells\ mL^{-1}$ | [66] |
| $p$ | Viral production rate | $(HIV\ RNA\ copies)\ day^{-1}$ | [49] |
| $d$ | Maximum effector cell exhaustion rate | $day^{-1}$ | [83] |
| $K_D$ | Density of infected cells needed to reach a half-maximal effector cell exhaustion rate | $cells\ mL^{-1}$ | [66] |
| $\lambda_T$ | Production rate of target cells | $10^4\ cells\ mL^{-1}\ day^{-1}$ | [84] |
| $d_T$ | Death rate of target cells | $0.01\ day^{-1}$ | [68] |
| $f_L$ | Fraction of latently infected cells produced per infection | $10^{-6}$ | [49] |
| $a$ | Reactivation rate of latently infected cells | $0.001\ day^{-1}$ | [49] |
| $\delta$ | Death rate of infected cells | $1\ day^{-1}$ | [85] |
| $d_E$ | Death rate of effector cells | $2\ day^{-1}$ | [83] |
| $L_0$ | Steady state size of the latent reservoir | $1\ cell\ mL^{-1}$ | [78,79] |
| $d_L$ | Decay rate of the latent reservoir | $0.004\ day^{-1}$ | [14,68] |
| $\rho$ | Latently infected cell proliferation rate, $d_L + a - \log(2)/t_{1/2}$ | $0.0045\ day^{-1}$ | [49,86] |
| $c$ | Clearance rate of virus | $23\ day^{-1}$ | [87] |

to the viral load data for all individuals simultaneously. We applied left censoring to data points under the limit of detection.

We assumed the fitting parameters follow a logit-normal distribution as we constrained the parameters to be in a fixed range. $\lambda_E$, $b$, $m$, and $d$ were constrained between 0 and 10 times the reference values used in Conway and Perelson (Table 1). Imposing bounds on $\lambda_E$, $b$, $m$, and $d$ is not necessary; however, this allows us to get the most consistent results across all models tested. We fit $\beta$ and constrained its range between $10^{-7.5}$ and $10^{-12}$ mL (HIV RNA copies)$^{-1}$ day$^{-1}$ to avoid an unrealistically high viral surge within 24 hours post-ATI due to high values of $\beta$. Parameter $p$ was constrained between 0 and 5000 per day. Parameters $K_B$ and $K_D$ were constrained between 0 and 100,000 cells per mL. Without this upper bound, $K_D$ values often tended to infinity. Note that although $K_D$ does not appear in Simplified Model 1, it does appear in other model variations that we examined (see Section A in S1 Text). No covariate was used during the initial fitting and comparison of all models. A covariate based on whether a participant is classified as PTC or NC was used later with the best fit model to determine the factors that can distinguish these two groups. Model comparison was done using the corrected Bayesian Information Criterion (BICc) [88] as reported by Monolix.

## Result

### Model fit and comparison

We fit the Conway and Perelson model and variations of it with different assumptions on the dynamics of latent and effector cells (see Section A in S1 Text). The best fit model was Simplified Model 1 described in Eq (2), where effector cell exhaustion is disregarded. Its best-fit to the data is shown in Fig 1. Comparisons of best fits for other models along with population and individual parameters are shown in Figs A-F and Tables A-M in S1 Text. The only difference between Simplified Model 1 and the Conway & Perelson model is the lack of an exhaustion term, which suggests exhaustion is not necessary to explain early viral rebound dynamics (within the first 52 weeks) seen in this set of PWH. This is further supported by the negligible effect of exhaustion estimated from the best fits for the Conway & Perelson model and Simplified Model 3, where the values for $d$ and $K_D$ result in a much smaller exhaustion effect compared to the reference values (Tables B and D in S1 Text). This does not imply that effector exhaustion is not relevant to HIV rebound dynamics as it is potentially important to explain the eventual loss of control as occurred in the subset of PTCs who have a truncated x-axis in Fig 1 and in the CHAMP study [25]. In addition, it is an important mechanism to explain the phenomenon where a PTC can have a high viral set point pre-ART and a lower viral set point post-ATI [49], and a variety of other dynamical phenomena [56,64,89].

Interestingly, the best fit parameter values for Simplified Model 1 with the exception of $K_B$ are close to the parameter values used in Conway and Perelson (compare Tables 1 and 2). This is also true for Simplified Model 1 even when we do not impose bounds on $\lambda_E$, $b$, and $m$. All models with effector cell dynamics fit the data relatively well (Table A in S1 Text), which is indicative of the importance of including in the model the dynamics of a population of effector cells that can kill productively infected cells.

We stratified the best fit parameters (Simplified Model 1) for each person by whether they were PTC or NC (Fig H in S1 Text). The most statistically significant difference using the Mann-Whitney test between the PTC and NC group is the parameter $K_B$ ($p$-value: 0.0005), followed by the immune cell killing rate $m$ ($p$-value: 0.0169). All model parameters differ at least somewhat between these two groups; however, the degree of significance of the difference for each parameter varies among models. In contrast, the statistically significant difference of $K_B$ between PTC and NC is consistently observed across all models tested (Figs G-J in S1 Text).

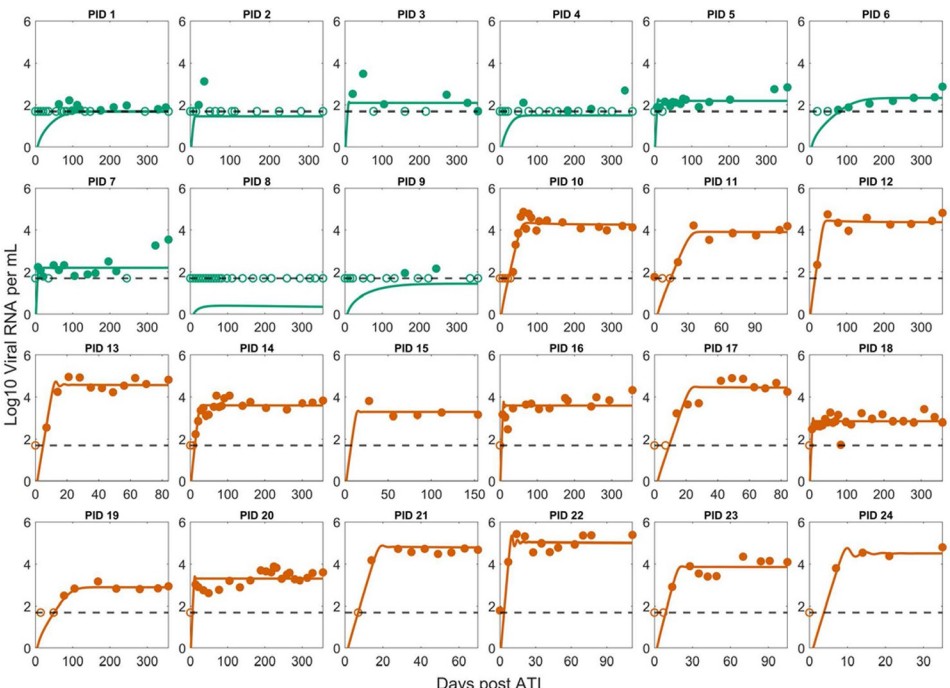

**Fig 1. Best fit of Simplified Model 1 to the post-ATI data from Sharaf et al. [57].** Green indicates PTC. Dark orange indicates NC. The horizontal dashed line is the limit of detection. Open circles are data points below the limit of detection (50 viral RNA copies/mL). Filled circles are data points above the limit of detection. Note the x-axis scale varies for the NCs as many of them were put back on ART terminating the data set.

This observation prompted us to fit Simplified Model 1 using a covariate on $K_B$ between PTC and NC. This modification significantly improves the model by 26 BICc points (Fig B and Table A in S1 Text); however, further inclusion of other covariates improves the fit slightly but worsens the BICc score (Table A in S1 Text). The best fit population value of $K_B$ for PTC 1.19 cell mL$^{-1}$ is consistent with the value in Conway and Perelson [49], which indicates a fast expansion of effector cells. The best fit population value of $K_B$ for NC is 230-fold higher, indicating a slower expansion of effector cells (Table 3).

Stratification of the best fit parameter values for Simplified Model 1 with a covariate on $K_B$ shows that the inclusion of a covariate on $K_B$ is sufficient to explain the variation between PTC and NC (Fig 2). Indeed, comparing the stratified individual parameters estimated using Simplified Model 1 without a covariate (Fig H in S1 Text) and with a covariate on $K_B$ (Fig 2), we find that the statistically significant difference between the PTC and NC observed for all other model parameters vanishes. For completeness, we also tested whether having a covariate on

**Table 2. Best fit population parameters (Simplified Model 1) vs. reference values from Conway and Perelson [49].**

| Parameter | Best fit values (population estimate) | Reference values |
|---|---|---|
| $\beta$ | $1.51 \times 10^{-8}$ mL day$^{-1}$ | $1.5 \times 10^{-8}$ mL day$^{-1}$ |
| $\lambda_E$ | 1.45 mL$^{-1}$ day$^{-1}$ | 1 mL$^{-1}$ day$^{-1}$ |
| $m$ | 0.83 mL day$^{-1}$ | 0.42 mL day$^{-1}$ |
| $b$ | 4.68 day$^{-1}$ | 1 day$^{-1}$ |
| $K_B$ | 52.62 mL$^{-1}$ | 0.1 mL$^{-1}$ |
| $p$ | 3477 day$^{-1}$ | 2000 day$^{-1}$ |

**Table 3. Best fit population parameters (Simplified Model 1 with a covariate on $K_B$) vs. reference values from Conway and Perelson [49].**

| Parameter | Best fit values (population estimate) | Reference values |
|---|---|---|
| $\beta$ | $1.58 \times 10^{-8}$ $mL$ (HIV RNA copies)$^{-1}$ $day^{-1}$ | $1.5 \times 10^{-8}$ $mL$ (HIV RNA copies)$^{-1}$ $day^{-1}$ |
| $\lambda_E$ | $0.50$ cells $mL^{-1}$ $day^{-1}$ | $1$ cells $mL^{-1}$ $day^{-1}$ |
| $m$ | $1.72$ $mL$ cell$^{-1}$ $day^{-1}$ | $0.42$ $mL$ cell$^{-1}$ $day^{-1}$ |
| $b$ | $5.40$ $day^{-1}$ | $1$ $day^{-1}$ |
| $K_B(PTC)$ | $1.19$ cells $mL^{-1}$ | $0.1$ cells $mL^{-1}$ |
| $K_B(NC)$ | $277$ cells $mL^{-1}$ | — |
| $p$ | $3220$ (HIV RNA copies) $day^{-1}$ | $2000$ (HIV RNA copies) $day^{-1}$ |

other parameters can improve the fit and remove the statistically significant variation in all other parameters including $K_B$. We find that adding a covariate on other parameters (one by one) not only does not improve the fitting compared to the fit with a covariate on $K_B$ (Table A in S1 Text), but it also has no effect on the statistically significant difference in $K_B$ between PTC and NC. To see whether the model can fit the data well without accounting for individual variation in $K_B$, we tested Simplified Model 1 without random effect for $K_B$, which results in worse fits (Table A in S1 Text). Altogether, these results support the inclusion of a covariate on $K_B$ as well as including individual variation in $K_B$.

## Analytical approximation of the rebound dynamics

To better explain why $K_B$ separates the NC and PTC participants, we approximated analytically the viral set point $V_{ss}$ after rebound (see Section E in S1 Text) and found

$$V_{SS} \approx \frac{p}{c} \frac{\left[ m\lambda_E - \left( (1-f_L)\beta\left(\frac{p}{c}\right)\left(\frac{\lambda_T}{d_T}\right) - \delta \right) d_E \right] K_B}{\left( (1-f_L)\beta\left(\frac{p}{c}\right)\left(\frac{\lambda_T}{d_T}\right) - \delta \right)(d_E - b) - m\lambda_E}. \tag{3}$$

Comparing $V_{ss}$ with the predicted viral load set-point obtained with the full model dynamics (Fig K in S1 Text) demonstrates the accuracy of this approximation.

The clinical classification of PTC individuals by Sharaf et al. is centered on the viral set point and how long it takes to rebound above the threshold of 400 viral RNA copies/mL (i.e., 24 weeks or longer of viral load below 400 copies/mL) [57]. The viral set point approximation $V_{ss}$ shows that the rebound classification is influenced by many factors in highly nonlinear ways. Yet, $V_{ss}$ is simply proportional to $K_B$, which may explain why the estimated value of $K_B$ for each individual separates PTC and NC in the model fits (Table H in S1 Text).

## Distinct immune response between PTC and NC

We calculated the effector cell expansion rate per infected cell $\left( \frac{bE}{K_B+I} \right)$ and the ratio of effector cells to infected cells ($E/I$) using the best fit parameters for each individual given in Table I in S1 Text. There is a clear distinction in both quantities between PTC and NC (Fig 3). The effector cell expansion rate for individuals in the PTC group is several times higher than for individuals in the NC group (Fig 3A). This result is not surprising given that $K_B$ is the only statistically significant difference between the two groups, where higher $K_B$ for the NC group leads to a slower expansion rate of the effector cells. Additionally, in PTC individuals, the model predicts at least one effector cell per infected cell throughout the studied duration,

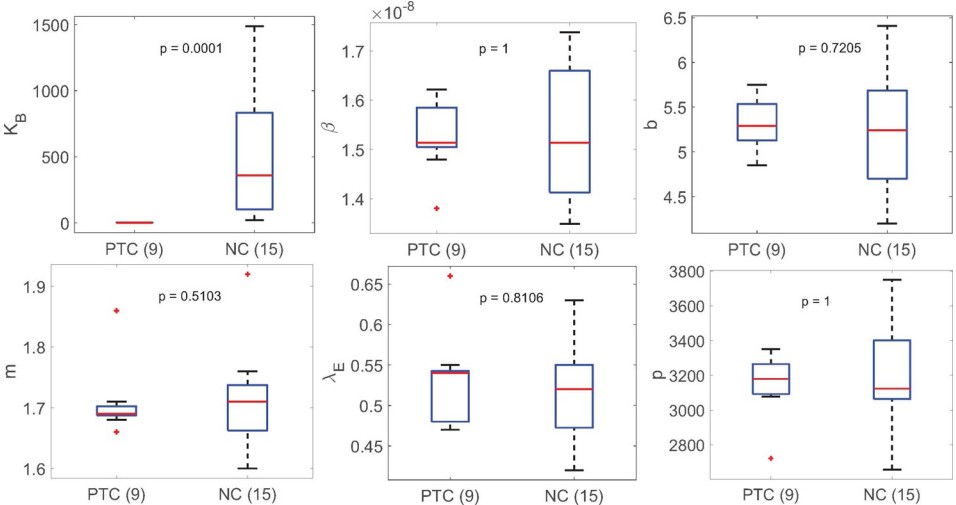

**Fig 2. Summary of best fit parameters in Simplified Model 1 with a covariate on $K_B$ stratified based on PTC or NC.** The median value of $K_B$ for PTC is 0.94 cell mL$^{-1}$. The difference in the values of $K_B$ is the most significant between PTC and NC. All other parameters are not statistically different between PTC and NC.

whereas in NC individuals, the model predicts about one effector cell for hundreds of infected cells on average (Fig 3B).

## Discussion

Differences in the size of the latent reservoir at the time of ART interruption and virus-immune interactions have been recognized as key factors in achieving post-treatment control [49]. Here, our modeling results robustly demonstrate that differences in the expansion rate of the effector cells differentiate PTC and NC in this cohort of 24 PWH.

The size of the latent reservoir determines the overall rate of reactivation of latent cells, which can lead to HIV-1 rebound post-ATI, as pointed out by Hill et al. [35]. However, to do so requires the reactivated cells to start a chain of infections that leads to exponential growth. If the immune system detects these newly reactivated cells and reacts fast enough and strongly enough, it can interrupt this chain of infection events and lead to PTC rather than non-control.

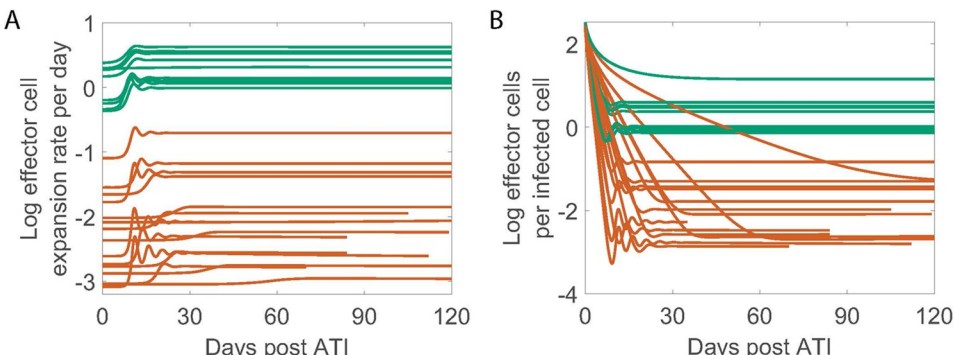

**Fig 3. Model predicted viral and immune dynamics using Simplified Model 1 with a covariate on $K_B$.** Green is PTC and dark orange is NC. (A) The effector cell expansion rate ($\frac{bE}{K_B+I}$) on a log10 scale. (B) The ratio of E vs. I plotted on a log10 scale.

Thus, in addition to the size of the reservoir, which Sharaf et al. [57] found to be 7-fold lower in PTCs than in the NCs, other factors, such as virus-immune interactions, may also affect the viral dynamics post-ATI. Esmaeilzadeh et al. [47] showed that autologous neutralizing antibodies apply selection pressure on rebounding variants and may play a role in mediating HIV suppression after ATI. Conway and Perelson [49] suggested that when the immune response is strong, PTC can be achieved. Our modeling results illustrate this concept by accurately capturing the early viral rebound dynamics in this cohort of 24 PWH.

We found that $K_B$ the density of infected cells needed to stimulate effectors cells into growth at their half-maximal rate, is significantly lower in PTC compared to NC. Since the clinical definition of PTC depends on the post-ATI set point viral load, and we showed there is an approximately linear relationship between this viral set point and $K_B$ in our model (Eq 3), this explains the distinct estimates of $K_B$ for PTCs and NCs. These results established an analytical relationship between an early and suppressive immune response and the viral rebound dynamics, echoing the recent results in the SIV system by Vemparala et al. [90]. In practice, it is unlikely that all model parameters including $K_B$ for an individual can be estimated prior to an ATI study. However, if an estimate of the value of $K_B$ can be obtained, perhaps via correlation with individual's immunological markers, then it may be possible to determine a threshold value of $K_B$ that separates PTCs and NCs a priori.

A plausible biological interpretation of the variation in $K_B$ comes from the studies of T cell expansion after contact with an antigen presenting cell (APC) [91,92]. Derivation of $K_B$ based on the binding kinetics of a T cell to an APC results in $K_B = \frac{k_p + k_d}{k_b}$, where $k_b$, $k_d$ and $k_p$ are the rate constants for T cells binding, dissociation and first order activation/proliferation, respectively. This means that a higher binding affinity, i.e., higher $k_b$ and/or lower $k_d$, would result in a lower $K_B$, which gives the interpretation that effector cells in PTC individuals have a higher binding affinity to the antigens presented on the surface of HIV-1 infected cells than the effector cells in NC individuals. This implies that the observed variation in $K_B$ may relate to different class I HLA alleles that present the HIV-1 peptides to CD8+ T cells carried by PTC and NC [93–98]. Alternatively, $K_B$ could also vary with the effector cell proliferation rate and cytotoxicity. For example, antigenic stimulation in vitro and ex vivo experiments have shown that CD8 + T cells from controllers can proliferate and increase their cytolytic capability more efficiently than those from progressors [99–101]. The effector cell population also includes other cell types, such as NK cells, and there is some evidence that NK cells may play an earlier role than CD8+ T-cells in the control of HIV-1 [42,48]. However, these interpretations remain speculations as $K_B$ is modeled as the number of infected cells needed to stimulate robust effector cell expansion, so the precise mechanism associated with $K_B$ is difficult to infer without a more detailed immunological model and data. Delineating the exact biological mechanisms that contribute to the differentiating values of $K_B$, perhaps via ex-vivo stimulation, characterization, and TCR-sequencing of CD8+ effector cells in addition to viral sequencing from individuals participating in ATI trials, may aid in finding the right targets for an effective design of vaccines and treatments for HIV-1.

Our results demonstrate that a variation of the Conway-Perelson model without effector cell exhaustion can characterize the viral rebound dynamics in the cohort of 24 individuals from Sharaf et al. [57]. However, there are limitations to the current study that should be noted for future investigation. First, although there is considerable heterogeneity in viral rebound dynamics in the 24 studied participants, this data set is still limited and may not be representative of the broader population of PWH. For instance, the CHAMP study [25] contains individuals with more complex viral rebound dynamics and sometimes a very high initial viral peak before viral remission, which our model fits miss. In addition, since we employed a

population fitting approach to fit a data set with mostly non-oscillatory individual data, the best fit model fails to capture the oscillatory features exhibited by several individuals (13, 20, 22, and 23 –Figs 1 and B in S1 Text). This suggests further examination of our modeling framework using data with more heterogeneous viral rebound dynamics is necessary to generalize our results.

Second, while our parameter estimates indicate limited exhaustion of effector cells (Tables B and D in S1 Text), this could be due to several factors such as the duration of ART and the one-year time constraint we put on the data. ART has been shown to restore the functionality of HIV-1 specific CD8+ T-cells and NK cells [102,103], hence lowering the exhaustion level of effector cells. Therefore, exhaustion at the start of ATI may potentially be very limited and unobservable to our model within the one-year time constraint. This limitation is evident when considers participants such as PTC number 7, who showed signs of viral rebound (~2 log increase in viral load) toward the end of year one, which is not captured by the best fit models. Thus, it is likely that effector cell exhaustion plays a role in PTCs who rebound later. Future studies should look further into the interaction of the immune system and virus over longer time scales to examine the possibility of rebound after the initial short-term viral remission. It is also worth pointing out that Simplified Model 2, which does not consider the dynamics of latently infected cells, can fit the data almost as well as Simplified Model 1. This is also likely due to the maximum one-year time constraint in the data and the slow dynamics of the latent reservoir. When considering a time scale on the order of years, some of the assumptions we made in order to derive the approximation of the viral set point are unlikely to hold such as that target cells remain at their disease-free equilibrium value of $\lambda/d_T$–see the derivation in S5. Thus, this necessitates more intricate mathematical analyses to ascertain the impact of immune parameters, such as $K_B$, on the viral set point [104].

A third - and intriguing–observation, which we did not consider, is that in some PWH, the viral set point post-ATI is orders of magnitude lower than before ART [22]. Qualitatively, this relates to the bistability of the Conway-Perelson model with respect to different initial conditions such as the size of the latent reservoir and the immune response at the time of ATI. If ART can sufficiently reduce the viral burden, it may allow the immune response to catch up. This may happen via the restoration of functionality for HIV-1 specific CD8+ T-cells and NK cells during ART [102,103]; however, the restoration effect may not be sufficient to improve the immune response in NCs to the level of PTCs in most cases [105].

Lastly, by choosing to focus on estimating the immune response parameters in our study, we assumed that the initial conditions of the model for all 24 individuals at the time of ATI were the same. While this assumption is not ideal, when fitting both the initial conditions and the immune parameters, the resulting parameter estimates vary significantly across all participants and models, likely due to parameter identifiability issues, which makes determining the key parameters an unrealistic task. Nevertheless, even with these limitations, our modeling results robustly demonstrate that variation in the effector cell expansion rate sufficiently captures the heterogeneity in viral rebound dynamics post-ATI.

## Supporting information

**S1 Text. Supplementary Material.** Table A. Model fit comparison (Monolix 2023R1, Lixoft, SA, Antony, France). There are four categories separated by double-lines. Bolded values are the lowest values in the category. The first category contains the Conway and Perelson model with five variations without covariates. The second category tests single covariates for the best fit model in the first category. The third category shows two examples of using two covariates that produce the lowest fitting error. The last category tests the Simplified Model 1 without

random effect for $K_B$. Without a random effect, the population estimate for $K_B$ is 16.88 (S.E. 0.12)cells mL$^{-1}$. Table B. Best-fit population parameters for the Conway & Perelson Model vs. reference values from Conway and Perelson [49]. Note the estimated values of $K_D$ and d result in a negligible effect of exhaustion. Table C. Best-fit population parameters for the Simplified Model 2 vs. reference values from Conway and Perelson [49]. Table D. Best-fit population parameters for the Simplified Model 3 vs. reference values from Conway and Perelson [49]. Note that the estimated values of $K_D$ and $d$ together imply the exhaustion effect is very small, which may be because of this particular set of participants or that it cannot be observed from the limited data. Table E. Best-fit population parameters for the Simplified Model 4 vs. reference values from Conway and Perelson [49]. Note that $m^*$ is not the same as $m$. Table F. Best-fit population parameters for the Simplified Model 5 vs. reference values from Conway and Perelson [49]. Note that $m^*$ is not the same as $m$. Table G. Individual best-fit parameters–Conway & Perelson model. Table H. Individual best-fit parameters–Simplified Model 1. Table I. Individual best-fit parameters–Simplified Model 1 with a covariate on $K_B$ (1 is PTC and 2 is NC). For $K_B$, the mean and SD (%) are reported for individual group PTC/NC. Table J. Individual best-fit parameters–Simplified Model 2. Table K. Individual best-fit parameters–Simplified Model 3. Table L. Individual best-fit parameters–Simplified Model 4. Table M. Individual best-fit parameters–Simplified Model 5. Fig A. Best fit of the Conway & Perelson model to the post-ATI data from Sharaf et al. [57]. Green indicates PTC. Dark orange indicates NC. The horizontal dashed line is the limit of detection. Open circles are data points below the limit of detection (50 viral RNA copies/mL). Filled circles are data points above the limit of detection. Fig B. Best fit of Simplified Model 1 with a covariate on $K_B$ to the post-ATI data from Sharaf et al. [57]. Lines show the model predictions using the best fit parameter values from Simplified Model 1 with a covariate on $K_B$. Green indicates PTC. Dark orange indicates NC. The horizontal dashed line is the limit of detection (50 viral RNA copies/mL). Open circles are data points below the limit of detection. Filled circles are data points above the limit of detection. Fig C. Best fit of the Simplified Model 2 to the post-ATI data from Sharaf et al. [57]. Green indicates PTC. Dark orange indicates NC. The horizontal dashed line is the limit of detection. Open circles are data points below the limit of detection (50 viral RNA copies/mL). Filled circles are data points above the limit of detection. Fig D. Best fit of the Simplified Model 3 to the post-ATI data from Sharaf et al. [57]. Green indicates PTC. Dark orange indicates NC. The horizontal dashed line is the limit of detection. Open circles are data points below the limit of detection (50 viral RNA copies/mL). Filled circles are data points above the limit of detection. Fig E. Best fit of the Simplified Model 4 to the post-ATI data from Sharaf et al. [57]. Green indicates PTC. Dark orange indicates NC. The horizontal dashed line is the limit of detection. Open circles are data points below the limit of detection (50 viral RNA copies/mL). Filled circles are data points above the limit of detection. Fig F. Best fit of the Simplified Model 5 to the post-ATI data from Sharaf et al. [57]. Green indicates PTC. Dark orange indicates NC. The horizontal dashed line is the limit of detection. Open circles are data points below the limit of detection (50 viral RNA copies/mL). Filled circles are data points above the limit of detection. Fig G. Summary of best-fit parameters in the Conway & Perelson Model stratified based on PTC or NC. The difference in the values of $K_B$ is the most significant between PTC and NC. Fig H. Summary of best fit parameters in Simplified Model 1 stratified based on PTC or NC. The difference in the values of $K_B$ is the most significant between PTC and NC. Fig I. Summary of best-fit parameters in the Simplified Model 2 stratified based on PTC or NC. The difference in the values of $K_B$ is the most significant between PTC and NC. Fig J. Summary of best-fit parameters in the Simplified Model 3 stratified based on PTC or NC. The difference in the values of $K_B$ is the most significant between PTC and NC. Fig K. Approximation of the set point viral load $V_{ss}$ using the best fit parameters for each participant compared with the numerical

solution of Simplified Model 1 ($Eq 2$). The horizontal black dotted line is the approximation $V_{ss}$ ($Eq 3$). Green and dark orange curves correspond to the viral load of the PTC and NC participants, respectively, as predicted by the Simplified Model 1. The horizontal pink dashed line is the 400 viral RNA copies/mL threshold used in the classification of PTC. For participant number 8, $V_{ss}$ is at 0.61 viral RNA copies/mL, which is the full model predicted viral set point after day ~12000.
(DOCX)

**S1 Data. This contains the de-identified viral load data of all 24 participants analyzed in this study.**
(CSV)

## Acknowledgments

We would like to thank Christiaan van Dorp for his insightful comments. We thank the participants, staff, and investigators of the ACTG studies.

**Disclaimer**

This report was prepared as an account of work sponsored by an agency of the United States Government. Neither the United States Government nor any agency thereof, nor any of their employees, makes any warranty, express or implied, or assumes any legal liability or responsibility for the accuracy, completeness, or usefulness of any information, apparatus, product, or process disclosed, or represents that its use would not infringe privately owned rights. Reference herein to any specific commercial product, process, or service by trade name, trademark, manufacturer, or otherwise does not necessarily constitute or imply its endorsement, recommendation, or favoring by the United States Government or any agency thereof. The views and opinions of authors expressed herein do not necessarily state or reflect those of the United States Government or any agency thereof.

## Author Contributions

**Conceptualization:** Tin Phan, Jessica M. Conway, Jonathan Z. Li, Ruy M. Ribeiro, Ruian Ke, Alan S. Perelson.

**Data curation:** Jonathan Z. Li, Ruian Ke, Alan S. Perelson.

**Formal analysis:** Tin Phan, Ruian Ke, Alan S. Perelson.

**Funding acquisition:** Jonathan Z. Li, Ruy M. Ribeiro, Ruian Ke, Alan S. Perelson.

**Investigation:** Tin Phan, Jessica M. Conway, Nicole Pagane, Jasmine Kreig, Narmada Sambaturu, Sarafa Iyaniwura, Ruy M. Ribeiro, Ruian Ke, Alan S. Perelson.

**Methodology:** Tin Phan, Jessica M. Conway, Ruy M. Ribeiro, Ruian Ke, Alan S. Perelson.

**Project administration:** Ruy M. Ribeiro, Ruian Ke, Alan S. Perelson.

**Resources:** Jonathan Z. Li, Ruy M. Ribeiro, Ruian Ke, Alan S. Perelson.

**Software:** Tin Phan, Ruian Ke, Alan S. Perelson.

**Supervision:** Jonathan Z. Li, Ruy M. Ribeiro, Ruian Ke, Alan S. Perelson.

**Validation:** Tin Phan, Jessica M. Conway, Nicole Pagane, Jasmine Kreig, Narmada Sambaturu, Sarafa Iyaniwura, Alan S. Perelson.

**Visualization:** Tin Phan.

**Writing – original draft:** Tin Phan, Nicole Pagane.

**Writing – review & editing:** Tin Phan, Jessica M. Conway, Nicole Pagane, Jasmine Kreig, Narmada Sambaturu, Sarafa Iyaniwura, Jonathan Z. Li, Ruy M. Ribeiro, Ruian Ke, Alan S. Perelson.

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
