## [Decision Letter · Decision Letter 0]

14 Jun 2024

Dear Dr. Perelson,

Thank you very much for submitting your manuscript "Understanding early HIV-1 rebound dynamics following antiretroviral therapy interruption: The importance of effector cell expansion" for consideration at PLOS Pathogens. As with all papers reviewed by the journal, your manuscript was reviewed by members of the editorial board and by several independent reviewers. The reviewers appreciated the attention to an important topic. Based on the reviews, we are likely to accept this manuscript for publication, providing that you modify the manuscript according to the review recommendations.

Sincerely,

Daniel C. Douek

Academic Editor

PLOS Pathogens

Richard Koup

Section Editor

PLOS Pathogens

Michael Malim

Editor-in-Chief

PLOS Pathogens

orcid.org/0000-0002-7699-2064

Reviewer Comments (if any, and for reference):

Reviewer's Responses to Questions

**Part I - Summary**

Reviewer #1: The manuscript is well-written with sufficient reference to the existing literature on HIV latency and the mathematical modeling thereof. The primary finding is that both the dynamics of viral rebound following the interruption of ART and the variation observed between post treatment controllers and noncontrollers are captured by a model with simplified effector cell dynamics (no exhaustion). Further, the distinction between PTC and NC can be made with a single parameter (K_B).

Models of viral infection that include immune cells come in many varieties, at many levels of complexity, and they all have many parameters. To my eyes, the significance of the results presented here are not necessarily in the specific mechanisms suggested (which will require much further experimental work to either confirm or reject) but the fact that a relatively simple model is able to capture the dynamics. As a result, future complex models for viral rebound will need to justify the level of complexity they contain, or acknowledge that the dynamic of rebound are already explained by a simpler submodel.

The general execution of this manuscript is excellent, and I have only minor comments to add prior to its acceptance.

Also, the supplementary materials provided are extremely helpful.

Reviewer #2: The paper studies the complex dynamics of HIV-1 rebound post-ART interruption. The authors developed a dynamic model based on the theoretical framework proposed by Conway and Perelson, which considers virus-immune interactions to understand viral rebound dynamics. The model is evaluated using viral load data from 24 individuals, highlighting the significance of the effector cell expansion rate in distinguishing post-treatment controllers (PTC) from non-controllers (NC).

**Part II – Major Issues: Key Experiments Required for Acceptance**

Reviewer #1: N/A

Reviewer #2: I do not find major issues.

**Part III – Minor Issues: Editorial and Data Presentation Modifications**

Reviewer #1: In describing the original Conway & Perelson model (l.98-l.124) care is sometimes taken about whether rates are constant or per capita constant, but at other times the distinction is not made. For example, d_T should be per capita, while it is unclear what the proper label for m would be. These discrepancies should be rectified, or should the could simply be referred to as "rates".

When discussing data fitting (l.163-l.178) it is slightly confusing to read references to parameters, such as K_D, that do not appear in the main model.

In Figure 1, the x-axes should all be the same. It is good that the existence of different axes is mentioned in the text, but I do not feel that the gain in ease of seeing individual data points outweighs the loss in visibility of dynamic rates.

Relatedly, the reason that some patients datasets are truncated (resumption of ART) is mentioned only in the caption of Figure 1, and should be mentioned in the primary text.

In the supplementary materials, the BICc for Simplified Model 2 is very close to that of Simplified Model 1. It seems worth a mention in the main text that fits of similar quality are achieve with a model that includes no dynamics in the latent cell class.

Reviewer #2: The study is based on data from 24 individuals, which may not be representative of the broader population of people living with HIV-1. The authors may want to add this limitation to the discussion section.

While the model finds limited impact of effector cell exhaustion in the short term, it does not fully explore its potential role in long-term viral dynamics and control. If these PTCs experience viral rebound later, does that mean effector cell exhaustion begins to play a role?

On page 4, lines 122-123, the authors mention homeostatic proliferation of latently infected cells. However, Ref 68 concerns asymmetric cell division. The following references discuss latency homeostasis: (1) PLoS Computational Biology 5 (10), e1000533, 2009; (2) Journal of Theoretical Biology 260 (2), 308-331, 2009.

PLOS authors have the option to publish the peer review history of their article (what does this mean?). If published, this will include your full peer review and any attached files.

Reviewer #1: No

Reviewer #2: No

Figure Files:

Data Requirements:

Please note that, as a condition of publication, PLOS' data policy requires that you make available all data used to draw the conclusions outlined in your manuscript. Data must be deposited in an appropriate repository, included within the body of the manuscript, or uploaded as supporting information. This includes all numerical values that were used to generate graphs, histograms etc.. For an example see here: http://www.plosbiology.org/article/info:doi%2F10.1371%2Fjournal.pbio.1001908#s5.

Reproducibility:

References:

---

## [Editor Report · Decision Letter 1]

27 Jun 2024

Dear Dr. Perelson,

We are pleased to inform you that your manuscript 'Understanding early HIV-1 rebound dynamics following antiretroviral therapy interruption: The importance of effector cell expansion.' has been provisionally accepted for publication in PLOS Pathogens.

Best regards,

Daniel C. Douek

Academic Editor

PLOS Pathogens

Richard Koup

Section Editor

PLOS Pathogens

Michael Malim

Editor-in-Chief

PLOS Pathogens

orcid.org/0000-0002-7699-2064
---

## [Editor Report · Acceptance letter]

24 Jul 2024

Dear Dr. Perelson,

We are delighted to inform you that your manuscript, "Understanding early HIV-1 rebound dynamics following antiretroviral therapy interruption: The importance of effector cell expansion.," has been formally accepted for publication in PLOS Pathogens.

Best regards,

Michael Malim

Editor-in-Chief

PLOS Pathogens

orcid.org/0000-0002-7699-2064